

# The role of long-term mineral and manure
# fertilization on P species accumulation and
# phosphate solubilizing microorganisms in paddy
# red soils
Shuiqing Chen[a], Jusheng Gao[b], Huaihai Chen[c], Zeyuan Zhang[a], Jing Huang[b], Lefu Lv[a],
Jinfang Tan[a], Xiaoqian Jiang[a]*
[a]School of Agriculture, Sun Yat-sen University, Guangzhou, Guangdong 510275, PR
China
[b]Qiyang Agro-ecosystem of National Field Experimental Station, Institute of
Agricultural Resources and Regional Planning, Chinese Academy of Agricultural
Sciences, Qiyang 426182, China
[c]School of Ecology, Sun Yat-sen University, Guangzhou, Guangdong 510275, PR China
*Corresponding to:
Xiaoqian Jiang
Mailing address: School of Agriculture, Sun Yat-sen University, Guangzhou 510275,
Guangdong, PR China
Tel: +86 18665976051



Email: jiangxq7@mail.sysu.edu.cn

## Abstract

Fertilization managements have important impacts on soil P transformation, turnover,
and bioavailability. Thus, long-term fertilization experiments (~38 years) with the
application of different inorganic and organic fertilizers in paddy red soils were
conducted to determine their effect on P pool accumulation and microbial communities,
especially for phosphate solubilizing microorganisms (PSM). Long-term inorganic P
fertilization increased the concentrations of total P (~479 mg/kg), available P (~417
mg/kg), and inorganic P (~18 mg/kg), but manure fertilization accelerated the
accumulation of organic P, especially for orthophosphate monoesters (e.g. myo-IHP,
~12 mg/kg). Long-term mineral fertilization decreased bacterial richness, evenness, and
complexation of bacterial networks. In contrast, long-term manure fertilization and
rhizosphere accumulated more amounts of total carbon, total nitrogen, and organic
carbon, as well as regulated the soil pH, thus improving the separation of bacterial
communities. Unlike bacteria, the responses of fungi to those factors were not sensitive.
Furthermore, PSM compositions were greatly influenced by fertilization managements





and rhizosphere. For example, inorganic P fertilization increased the abundance of
*Thiobacillus* (i.e. the most abundant phosphate solubilizing bacteria (PSB) in this study)
and shifted the community structure of PSB. Correspondingly, the concentrations of
inorganic and total P were the key factors for the variation of PSB community structure.
These findings are beneficial for understanding P accumulation, responses of PSB, and
soil P sustainable fertility under different fertilization strategies.
**Keywords**: long-term fertilization, P species accumulation, phosphate solubilizing
microorganisms, paddy red soils, P-NMR

# 1. Introduction

Phosphorus (P) as an essential nutrient for crop growth has been widely applied to soil
through mineral and/or organic fertilization (Grant et al., 2005). Manures have been
frequently used as organic fertilizers in agriculture production (Braos et al., 2020). The
P from manures exists in forms of various inorganic and organic species, whereas
mineral fertilizers usually only contain highly soluble $Ca(H_2PO_4)_2$ (Sharpley and Moyer,
2000). Fertilization managements are important factors for P species transformation and
bioavailability. For example, mineral fertilization results in an initial high P availability
but follows a decrease of P concentration over time by adsorption, complexation, and



precipitation with soil particles. On the other side, the application of manure usually
leads to an accumulation in labile organic P pools with potential supply to plants
(Schneider et al., 2016). Additionally, the application of mineral fertilizer and manure
brought different changes in soil physical, chemical, and biological attributes such as
soil pH, organic carbon, microbial communities, and so on, which also induce different
P transformation processes and potential availability (Yue et al., 2016; Tao et al., 2021).
Soil microorganisms are usually involved in a wide range of biological processes
including the transformation of insoluble soil nutrients (Babalola and Glick, 2012).
After long-term fertilization, insoluble or soluble organic matter in soil may increase,
thus leading to the increases of microbial biomass and activity (Marschner et al., 2003).
Among them, phosphorus solubilizing microorganisms (PSM) could solubilize
insoluble inorganic P, mineralize organic P, and play an important role in P
transformation and availability (Sharma et al., 2013). The response of PSM in soil is
strongly related to the availability of P which is greatly different under various
fertilization managements (Sánchez-Esteva et al., 2016; Gómez-Muñoz et al., 2018;
Raymond et al., 2021).
Currently, the information about how long-term various inorganic and organic



fertilization managements affect the evolution characteristics of different P pools
remains scarce. Furthermore, the responses of microbial community especially PSM
shift in bulk and rhizosphere soils to the different P pool evolution under various
fertilization managements are still unclear. This information plays a pivotal role for
understanding soil P transformation mechanisms and evaluating sustainable P fertility
and potential bioavailability in agriculture managements. The accumulation, turnover,
and bioavailability of soil P pool under different fertilization managements could be
well evaluated by long-term fertilization experiences. Currently, numerous long-term
fertilization experiences have been established to evaluate the impact of different
fertilizer amendments on crop production and at the same time provide valuable
information on soil fertility by investigating changes in soil process over time (Wen et
al., 2019). Thus, in this study, long-term fertilization experiments (~38 years) under
inorganic fertilizer and/or manure amendments were conducted to determine their
effects on P pool accumulation, soil microbial communities, and PSM in paddy red soils.
We hypothesized that the inputs of long-term mineral fertilizer and manure (1) caused
P accumulation with different species and potential availability and (2) drove the shift
of soil microbial community including PSM.



## 2.Materials and methods

### 2.1. Field design and sampling

Long-term fertilization experiments were conducted since 1982 in a national observation and research station of farmland ecosystem (26°45′N, 111°52′E), Qiyang, Hunan Province, China. Rice (*Oryza sativa*) is the major crop in this region. The soil was classified as Ferralic Cambisol according to World Reference Base for soil resources (Wrb, 2014), and classified as red soil according to Chinese soil classification (Baxter, 2007). The experimental field was disposed with five different fertilizer treatments: CK (control without fertilizer), NPK (mineral N, P, and K fertilizers), M (cattle manure), NPKM, and NKM (Qaswar et al., 2020). Mineral fertilizers were applied in the forms of urea for N, calcium superphosphate for P, and potassium chloride for K with the amounts of 145 kg ha$^{-1}$ of N, 49 kg ha$^{-1}$ of P, and 56 kg ha$^{-1}$ of K, respectively. Additionally, the manure was added with the average nutrient contents including 18000 kg ha$^{-1}$ of C, 145 kg ha$^{-1}$ of N, 49 kg ha$^{-1}$ of P, and 56 kg ha$^{-1}$ of K. All the mineral fertilizers and manure were applied as basal application. Bulk soil samples collection with five different fertilizer treatments (1-20 cm topsoils) were conducted before the harvest of late rice in October 2020 with field replications. Besides,



approximately 1 mm of soil on the rice roots was collected as rhizosphere soil (Shao et
al., 2021). Soil samples used for physical and chemical analyses were four replications
(2 field replication×2 replication of each field, n=4) and those for DNA extraction were
six replications (2 field replication×3 replication of each field, n=6).

## 2.2. Soil physical and chemical properties

Soil pH was measured by pH meter in the mixed solution (the mass ratio of soil and
water is 1:2.5). Soil moist content was measured by drying moist soil to constant mass
at 105 °C. Total carbon (TC), organic carbon (OC), and total nitrogen (TN) were
determined by CHNS elemental analyzer (Vario EL Cube manufactured by Elementar,
Germany) (Schumacher, 2002). The soil extracts with 2 M KCl treatment were
determined for ammonia-N ($NH_4^+$) by indophenol blue colorimetric method (Dorich
and Nelson, 1983), and for nitrate-N ($NO_3^-$) by dual-wavelength ultraviolet
spectrophotometry (Norman et al., 1985). After potassium persulfate and $H_2SO_4$ pre-
digestion (Bowman, 1989), soil samples were determined for total P by a colorimetric
method (Murphy and Riley, 1962). The extraction of available phosphorus (AP) was
referred to the method described by Olsen (Olsen, 1954), and the concentration was
measured using a colorimetric method (Murphy and Riley, 1962).



The extracted P with 0.5 M NaHCO$_3$ before/after 24 h of CHCl$_3$ fumigation was
determined using ICP-OES (PerkinElmer, Avio 500, USA). A KEC factor of 0.4 was
used for the calculation of soil microbial biomass P. Soil microbial biomass P was
measured using a chloroform fumigation-extraction technique (Brookes et al., 1982).
Additionally, the activities of acid and alkaline phosphatase were assayed by the method
described by Tabatabai and Bremner (1969) using *p*-nitrophenyl phosphate as substrate
at 37 °C.

## 2.3 Organic P analyses

Soil organic P was extracted with NaOH-EDTA solution according to the method
described by Jiang et al. (2017). In short, 4 g air-dried soil was extracted for 4 h using
40 ml solution containing 0.25 M NaOH and 0.05 M Na$_2$EDTA. After centrifuging at
13,000 × *g* for 20 min, 2 mL aliquot of each supernatant was used to determine Fe, Mn,
and P by ICP-OES. The remaining supernatants were freeze-dried and prepared for
solution $^{31}$P-NMR spectroscopy. Each freeze-dried extract (~100 mg) was re-dissolved
in 0.1 mL of deuterium oxide and 0.9 mL of a solution containing 1.0 M NaOH and 0.1
M Na$_2$EDTA, then immediately determined with solution $^{31}$P-NMR spectra using a
Bruker 500-MHz spectrometer. The NMR parameters were: 28 K data points, 0.68 s



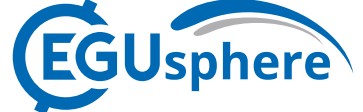

acquisition time, 90° pulse width, and 8000 scans. The repetition delay time was
calculated based on the concentration ratio of P to (Fe+Mn) according to the research
by Mcdowell et al. (2006). Peak areas were calculated by integration on spectra
processed with 2 and 7 Hz line-broadening using MestReNova software. Phosphorus
species were identified based on their chemical shifts, including orthophosphate (6
ppm), pyrophosphate (~ -5 ppm), polyphosphate (-4 to -5, -5 to -50 ppm),
orthophosphate monoesters (3 to 6, 6 to 7 ppm), orthophosphate diesters (3 to -4 ppm),
and phosphonates (7 to 50 ppm). The orthophosphate peak was standardized to 6 ppm
during processing (Cade-Menun et al., 2010; Young et al., 2013). Individual P
compounds were identified based on their chemical shifts from the study by (Cade-
Menun, 2015) and by spiking selected samples with myo-inositol hexakisohosphate
(myo-IHP), α- and β-glycerophosphates (Fig. S1 and S2).
The concentrations of individual P species were calculated by multiplying [31]P-NMR
proportions by the total NaOH-Na$_2$EDTA extractable P concentration. The α- and β-
glycerophosphates    and    mononucleotides    were    considered    as    degradation    of
orthophosphate diesters, though they were detected in the orthophosphate monoester
region (Young et al., 2013; Liu et al., 2015).



## 2.4. Soil DNA extraction, PCR amplification, Illumina Miseq sequencing, and bioinformatics analyses

The DNA was extracted from 0.25 g soil using FastDNA® Spin Kit (MP Biomedicals, USA). The purity and concentration of DNA were measured by Nanodrop 2000 (Thermo Fisher Scientific, USA). For bacteria, the V3-V4 region of the 16S rRNA gene was amplified with the primer pair 338F (5′-ACTCCTACGGGAGGCAGCAG-3′) and 806R (5′-GGACTACHVGGGTWTCTAAT-3′) (Caporaso et al., 2012; Dennis et al., 2013). For fungi, the primer pair ITS1F (CTTGGTCATTTAGAGGAAGTAA) and ITS2 (GCTGCGTTCTTCATCGATGC) were used to target the ITS1 region (Blaalid et al., 2013). After sequencing, the raw sequences of each sample were assembled by QIIME 2 according to the unique barcode after removing the adaptors and primer sequences (Bolyen et al., 2019). Demultiplexed sequences were quality filtered, trimmed, de-noised, and merged, then the QIIME2 dada2 plugin was used to identify and remove chimeric sequences to obtain the feature table of amplicon sequence variant (ASV) (Callahan et al., 2016). ASV sequences were aligned to the GREENGENES database and UNITE database separately to generate the taxonomy table for bacteria and fungi (Bokulich et al., 2018). Besides, phosphorus-solubilizing microbes were





collected according to the researches by RodríGuez and Fraga (1999) as well as  Alori
et al. (2017) ( see Table S1). The raw reads of bacteria and fungi were deposited in the
NCBI  Sequence  Read  Archive  (SRA)  database  under  accession  numbers
PRJNA804681 and PRJNA805018, respectively.
**2.5.  Statistical analyses**
All  statistical  analyses  were  conducted  using  SPSS  25.0.  All  indicators  between
different fertilizer treatments (i.e., CK, NPK, M, NPKM, and NKM) were tested for
significant differences (set to $p < 0.05$) by one-way ANOVA. The LSD was used to test
significant differences of all indicators between bulk and rhizosphere soils. Alpha (α)
diversity indices, such as Chao1 richness estimator and Shannon diversity index, were
calculated using the core-diversity plugin within QIIME2. Nonmetric multidimensional
scaling (NMDS) based on Bray Curtis distance was measured by R package "vegan"
and  visualized  via  R  package  "ggplot 2".  Co-occurrence  network  analysis  was
performed by using R package "psych" to calculate Spearman's rank correlations for
taxa among 6 repetitions of each treatment group and then Gephi 0.9.2 software was
used to draw networks. Redundancy analysis (RDA) was performed by Monte Carlo
analysis using Canoco 5 to reveal the association of microbial communities and soil





environmental factors.

## 3. Results

### 3.1 Soil physicochemical properties

In this study, we found that TC, TN, and OC increased significantly after the long-term
application of fertilizers, especially for manure fertilization (Table 1). It was expected
that the application of fertilizers increased the plant biomass such as plant residues and
root exudates (Tong et al., 2019). In addition, the input of manure also brought high C
and N contents in soil (Wei et al., 2017). The concentrations of microbial biomass P
increased under long-term fertilization (Table 1). Additionally, the activities of acidic
phosphatase (ACP) were higher than alkaline phosphatase activities (ALP) for all the
treatments (Fig. 1H and I). On the other side, soil pH value, gravimetric moisture, $NO_3^-$
-N, and $NH_4^+$-N contents were not affected by the long-term fertilizer treatments
significantly (Table 1). Notably, a previous study of the fields has found that the pH
decreased with NPK treatment but increased with organic fertilization (Ahmed et al.,
2019), which was inconsistent with this study. The possible reason is that two sampling
time was different and continuous heavy rainfall before sampling may also reduce the
difference of pH among treatments in this study.



## 3.2 Soil P species

Long-term application of inorganic P fertilizer (NPK and NPKM) could significantly

increase total P (TP), available P (AP) and inorganic P (IP) concentrations in both bulk

and rhizosphere soil (Fig. 1A, B, and C). The concentrations of NaOH-Na$_2$EDTA

extracted P in the soils were ~243-739 mg/kg, accounting for ~38-66% of total P (Table

2). Orthophosphate, pyrophosphate, orthophosphate monoesters (e.g. myo-IHP, scyllo-

IHP), and orthophosphate diesters (e.g. DNA) were found in the soils (Table 2). The

amounts of soil organic P (i.e. sum of orthophosphate monoesters and diesters) were

not much and accounted for 8-30% of total P (data not shown). Generally, the

concentrations of organic P were higher with long-term manure fertilization compared

to those of CK and NPK (Fig. 1D). Among the OP, the amounts of orthophosphate

monoesters (57-96 mg/kg) were higher than those of orthophosphate diesters (34-65

mg/kg) (Table 2). The long-term manure amendments had an obvious effect on the

accumulation of orthophosphate monoesters: the concentrations of orthophosphate

monoesters and myo-IHP were higher significantly with manure fertilization (i.e. M,

NPKM, NKM) than those with other treatments (i.e. CK, NPK) (Fig. 1E and G).

Phosphate monoesters were regarded as relatively stable and were the dominant group





of organic phosphorus compounds in most soils (Tabatabai, 1989), mainly including
inositol phosphates (e.g. myo, scyllo, D-chiro, neo) (Cosgrove and Irving, 1980; Turner
et al., 2002). The concentrations of orthophosphate diesters were also higher with
manure treatments compared to CK and NPK although the tendency was not significant
(Fig. 1F).
**3.3 Long-term fertilization and rhizosphere effect on the composition**
**of microbial community**
The dominant bacteria for different treatments at the phylum level were *Proteobacteria,*
*Acidobacteria, Chloroflexi,* and *Nitrospirae* and the dominant fungi were *Ascomycota*
and *Basidiomycota* (Fig. 2). As the most abundant phylum of bacteria, *Proteobacteria*
were     further     classified     into     *Alphaproteobacteria,     Betaproteobacteria,*
*Gammaproteobacteria, Deltaproteobacteria, Epsilonproteobacteria,* and unclassed
groups at the class level. *Gammaproteobacteria* was significantly more abundant for
manure   treatments   than   for   CK   and   NPK   treatments.   The   abundance   of
*Epsilonproteobacteria* increased after mineral fertilization. Both inorganic and organic
fertilization could increase the abundance of *Alphaproteobacteria* (Fig. S3). On the
other side, certain bacteria and fungi at the phylum level affected by fertilization were

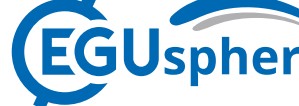

different in rhizosphere and non-rhizosphere soils. For example, the long-term manure
fertilization accumulated more *Spirochaetes* but less *Actinobacteria* and *TM7* in non-
rhizosphere soils (Fig. 2A and C). The relative abundance of *Ascomycota* increased
significantly with fertilization in non-rhizosphere soils but not the case in the
rhizosphere soils (Fig. 2B and D). These results suggested that both long-term
fertilization and rhizosphere affected the microbial community composition together.
The relative abundances of PSB were also greatly influenced by fertilization and
rhizosphere. The *Thiobacillus* was the most abundant bacterium at genus level and
increased with long-term input of inorganic P in both bulk and rhizosphere soils (Fig. 3
A and C). Additionally, the long-term manure fertilization increased the abundance of
*Flavobacterium* in bulk soil. On the other side, the *Fusarium* was the most abundant
fungus at genus level (Fig. 3 B and D). The influence of fertilization on the phosphorus-
solubilizing fungi (PSF) in bulk soil was not obvious. However, manure fertilization
increased the abundance of *Aspergillus* and *Trichoderma* in rhizosphere soils.
**3.4 Microbial community diversity**
Soil with long-term mineral fertilization (NPK) presented a lower bacterial richness and
evenness (i.e. Chao 1 and Shannon index) than those with manure fertilization



(M/NPKM/NKM) and even lower than control soil (CK), indicating that bacterial α-
diversity decreased after long-term mineral fertilizer regimes, but was not changed
under manure fertilization (Fig. 4). Other long-term field studies have also shown the
similar tendency (Li et al., 2015; Francioli et al., 2016; Wang et al., 2018). On the other
side, rhizosphere effect was clearly observed on the bacterial diversity: the richness and
evenness of bacterial community in rhizosphere soil were significantly higher than
those in non-rhizosphere soil (P<0.001). It is worth noting that fertilization and
rhizosphere effect have no obvious influence on fungal richness and evenness. It
suggested that long-term fertilization and rhizosphere affected the richness and
evenness of bacterial and fungal communities differently.
The plot of N ·MDS identified the variations in microbial β-diversity between different
sites, with the response of bacterial β-diversity being greater than that of fungal β-
diversity (Fig. 5). Specially, the profiles of bacterial β-diversity with manure
fertilizations (M, NKPM, NKM) were clearly separated from that for CK soil (Fig. 5 A
and C). The analysis of similarities (ANOSIM) revealed that R values for rhizosphere
soils between different fertilization treatments were higher than those for bulk soils
(Table S2). Accordingly, the variations in bacterial β-diversity of rhizosphere soils with



manure fertilization were greater than that of bulk soils (Fig. 5 A and C). These results
indicated that manure fertilization and rhizosphere effect exacerbated the variation of
bacterial β-diversity.

**276  3.5 Co-occurrence networks**

The co-occurrence network was used to analyze the ecological relationship of both
bacterial and fungal communities under five fertilization treatments. After long-term
mineral fertilization (NPK), total edges, average degree, positive edges, and
positive/negative edges ratio (i.e. P/N ratio) of bacteria and fungi network decreased
(Fig. 6 and Table S3), indicating that long-term mineral fertilization increased the
stability of microbial network (e.g. lower P/N ratio) but decreased the complexity of
network (e.g. less total edges and lower average degree) (Tu et al., 2020; Olesen et al.,
2007; Hernandez et al., 2021). Meanwhile, long-term manure treatments (M, NPKM,
NKM) increased the negative connections of microorganisms, and also promoted  the
stability of network (Zhou et al., 2020). Additionally, the high P input (NPKM vs NKM)
brought a larger and more complex but less stable bacterial network (e.g. more total
nodes, edges, average degree, average clustering coefficient, average path length, and
less modularity). However, the opposite tendency was shown for fungus network (Fig.





6 and Table S3), indicating that the response of bacteria and fungi to the input of
inorganic P was different.

**3.6 Factors correlating with microbial community diversity**

Redundancy analysis (RDA) was conducted to determine the correlation of soil
properties with microbial community diversity in bulk and rhizosphere soils. The results
showed that TC (10.4%, F=4.4, P=0.03), soil pH value (10.3%, F=4.4, P=0.03), TN
(10.1%, F=4.3, P=0.03), and OC (9.2%, F=3.9, P=0.03) were significantly correlated
with bacterial community diversity (Fig. 7A, Table S4). On the other side, for the fungus,
the soil properties had extremely small explainations of <4.4% for the variation for
fungi community (Table S4).
The RDA was also performed to establish the linkages of soil properties with
community diversity of PSM. The soil properties together explained more than 55% of
the variation in PSB community structure and those correlated with PSB contained TP
(27.5%, F=14.4, P=0.03) and IP (26.6%, F=13.7, P=0.03) (Fig. 7C and Table S5). The
PSB was well separated by RDA1 (52.60 %) between the samples with inorganic P
application (i.e. NPK, NPKM) and without inorganic P application (i.e. CK, M, NKM)
(Fig. 7C). The 30.08% of the total variance in the PSF community could be explained





by the first and second axes (Fig. 7D).

## 4. Discussion

### 4.1. Long-term fertilization on soil P accumulation

Long-term organic P fertilization increased the utilization of P for crops compared to
inorganic P fertilization. The same amount of P was added to soil whatever inorganic
or organic fertilization but the total P of soil was significantly higher with mineral
fertilization compared to manure treatment, suggesting more P was retained in soil and
less P was utilized by crops under long-term mineral fertilization (Fig. 1A). Several
researchers have already reported that inorganic P was easily immobilized by clay
minerals and was dominantly associated with amorphous Fe/Al oxides compared to
crystalline Fe/Al oxides fractions in many soil types such as Sandy soils, Ultisols,
Luvisol, Ferralic Cambisol, and so on (Arai et al., 2005; Rick and Arai, 2011; Jiang et
al., 2015; Ahmed et al., 2019).   On the other side, it has been confirmed that the
application of manure usually leads to an increase in labile organic P pools, which are
protected from the process of adsorption on clay minerals and are readily available to
plants (Braos et al., 2020; Kashem et al., 2004). In this study, manure fertilization
increased microbial biomass P concentration and alkaline phosphatase activity

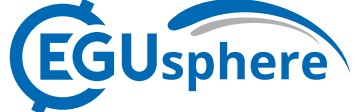

compared to mineral fertilization (Fig. 1I, Table 1). It was possible that the
mineralization of organic P such as orthophosphate diesters from microbes by alkaline
phosphatase increased under organic fertilization, thus improving the P availability for
crops.
The application of inorganic P fertilizer mainly increased the concentration of inorganic
P but manure fertilization accelerated the accumulation of organic P in soil (Fig. 1C and
D). Phosphorus speciation was usually regulated by the changes in soil mineralogy,
mineral and organic P inputs, biological production, and the utilization of various P
species (Turner et al., 2007; Jiang et al., 2017). Fertilization especially for manure
accelerated the accrual of organic carbon (Table 1) significantly, which also co-
accumulated organic P. Generally, the content of organic phosphorus (OP) from
manures accounts for a large proportion of total P, among which inositol phosphate
(IHP) was the most abundant OP (Maguire et al., 2004). Therefore, long-term manure
fertilization also increased the input of OP in the field. The organic P could be
effectively mineralized by microorganisms and thus transferred into various inorganic
P fractions (Song et al., 2007).
The application of manure increased the accumulation of orthophosphate monoesters



significantly, especially for myo-inositol phosphates (myo-IHP) (Fig. 1E and G).
Normally, phosphate monoesters were the main group of organic P compounds and
existed as IHP mainly in most soils (Turner et al., 2005). Those orthophosphate
monoesters were commonly stabilized by association with soil minerals such as Fe/Al
oxides (Celi and Barberis, 2007; Turner and Engelbrecht, 2011; Jiang et al., 2015).
Therefore, the stability and immobilization of orthophosphate monoesters promoted
their accumulation in soil no matter by the input of manure or by the P transformation.
On the other side, separate manure fertilization (M) also increased the contents of
orthophosphate diesters significantly (Fig. 1F). Long-term manure fertilization
accumulated more microbial biomass P significantly (Table 1) that were rich in
orthophosphate diesters (Turner et al., 2007). The accumulation of orthophosphate
diesters under manure fertilization was probably due to the reduced decomposition of
plant residues and manure or increased microbial synthesis under anaerobic paddy-rice
management (Jiang et al., 2017).
**4.2 Long-term fertilization and rhizosphere effect on soil microbial**
**communities**
Our results indicated that long-term mineral fertilization decreased bacterial richness,

evenness, and the complexation of bacterial networks. On the other side, long-term organic fertilization did not change the bacterial richness and evenness, and even promoted the separation of bacterial communities. Previous studies have reported that long-term mineral fertilization changed soil properties and these perturbations may have an adverse effect on soil microbes (Marschner et al., 2003; Geisseler and Scow, 2014; Liang et al., 2020). In contrast, organic fertilizer contained a large amount of organic matter which could be utilized by soil bacteria (Wu et al., 2020; Wu et al., 2021). Additionally, there were significant increases for diversity of bacterial communities in rhizosphere soil compared to bulk soil. Generally, microbes concentrated in the rhizosphere where organic compounds were released by plant roots (Achat et al., 2010), and plants tend to recruit bacteria as symbiotic microbes by releasing phenolic compounds (Gkarmiri et al., 2017; Badri et al., 2013).

Accordingly, redundancy analysis showed that the key factors related to the shift of bacterial communities included pH, TC, TN, and OC. The previous study showed that the soil bacteria community was indirectly impacted by pH via the alteration of metals and nutrient availability (Xiao et al., 2021), and directly modulated by the abundance and mineralization of carbon in soil (Chen et al., 2019) as well as soil nitrogen





deposition (Zeng et al., 2016). In this study, the long-term organic fertilization and
rhizosphere soil accumulated more TC, TN, and OC, which provided more nutrients,
changed the soil pH, and thus drove the shift of bacterial communities (Ingwersen et al.,
2008; Liu et al., 2019).
Additionally, the application of both mineral and organic fertilizers increased the
stability of bacterial networks (i.e., increasing negative correlations). Compared to CK,
long-term fertilization provided more nutrient elements, stimulated the growth and
competition of bacteria, and finally facilitated the stability of ecological network (Faust
and Raes, 2012; Simard et al., 2012).
It was worth noting that fertilization and rhizosphere effect had no obvious influence
on fungal community structure. Redundancy analysis showed that the explanations of
soil properties were extremely small for the variation for fungi community. It has been
found that fungi were less sensitive to soil substrates and environmental conditions
whereas bacteria were more sensitive (Dong et al., 2014). The high TOC provided by
the long-term fertilization and rhizosphere soil gave an advantage for bacteria to
compete with fungi for resources, thus decreasing influences of long-term fertilization
and rhizosphere on fungi (Zelezniak et al., 2015).





## 4.3 Response of PSM


*Thiobacillus* was the most abundant PSB at genus level and increased with the input of
inorganic P fertilizers in bulk and rhizosphere soil (Fig. 3 a and c). It was involved in
sulfur oxidation, and acidity resulted from sulfur oxidation could solubilize mineral P
(Aria et al., 2010). Acidic and anaerobic conditions provided by paddy-rice
management of red soil in this study were beneficial for the growth of *Thiobacillus*
considering that it belongs to acidophilic bacterium (Monachon et al., 2019; Kumar et
al., 2020). The applied calcium superphosphate as inorganic P fertilizer in this study
contained a certain amount of $CaSO_4$, therefore the input of inorganic P fertilizer also
provided S source for the growth of *Thiobacillus*. On the other side, *Fusarium* was the
most abundant PSF at genus level (Fig. 3 B and D) and was proven to produce organic
acid to solute the mineral P (Elias et al., 2016). It was known that *Fusarium* was widely
distributed in soil around the world and acted as a saprophyte (Deacon, 1997), among
which many species were also found as phytopathogens (Suga and Hyakumachi, 2004).
Besides, the long-term organic fertilization increased the abundance of *Flavobacterium,*
*Aspergillus*, and *Trichoderma. Flavobacterium* was associated with the degradation of
phosphotriester (Brown, 1980) and was proven to grow in a nutrient-rich condition

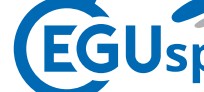

(Kraut-Cohen et al., 2021). *Aspergillus*, as a saprophytic fungus, could produce organic
acid to dissolve mineral phosphorus (Li et al., 2016) and also preferred to the nutrient-
rich condition (Martins et al., 2014). Additionally, *Trichoderma* as a biological control
fungi (Zin and Badaluddin, 2020) was colonized in the root epidermis and outer cortical
layers (Harman, 2006). Long-term organic fertilization provided more organic matter
for these microbes.
PSM could solubilize mineral P and mineralize organic P (Sharma et al., 2013). The
PSB of samples with inorganic P input (i.e. NPK, NPKM) and none mineral P
application (i.e. CK, M, NKM) could be well separated, indicating mineral P had a
strong effect on community diversity of PSB. Correspondingly, TP and IP were key
factors driving the diversity of soil PSB community and those indicators were all higher
significantly with inorganic P amendments (Fig. 1A, and C). As discussed before,
*Thiobacillus* as the most abundant PSB at genus level in this study increased with the
input of mineral P. It is because that mineral P could provide additional S source for the
growth of *Thiobacillus*. Furthermore, the availability of P in soil was considered as a
key condition for PSM to express P-solubilization traits. Low availability of P in soil is
widely considered as a favorable condition for PSM whereas recent studies suggested



that a minimum P threshold is required to achieve a response by plants (Sánchez-Esteva
et al., 2016; Gómez-Muñoz et al., 2018; Raymond et al., 2021).

## 5. Conclusion

Long-term inorganic and organic fertilization managements brought different effects on
P accumulation, microbial community, and PSB. Long-term mineral fertilization
increased inorganic and available P concentrations. In contrast, manure fertilization
increased soil organic P concentrations, microbial biomass P contents, and alkaline
phosphatase activity, which is beneficial for the mineralization of organic P, especially
for orthophosphate diesters.
The turnover of P by bacteria seems strong under long-term organic fertilization and
rhizosphere soil considering that more nutrient was provided for bacteria and the
bacterial community diversity increased. Furthermore, the responses of PSM to
different fertilization managements were also different. For example, inorganic P
fertilization increased the abundance of *Thiobacillus* (i.e. the most abundant PSB in
studied soil) whereas organic fertilization increased the abundance of *Flavobacterium,*
*Aspergillus*, and *Trichoderma*. The concentrations of TP and IP strongly influenced by
inorganic P fertilization were key factors driving the diversity of soil PSB community.



These findings provide useful insights into P accumulation, turnover, and soil P
sustainable fertility under different fertilization strategies.

**Acknowledgments:**
This study was financially supported by the National Natural Science Foundation of
China (No.41907063).
**Author contribution:**
Shuiqing Chen: Investigation, Data curation, Formal analysis, Writing- Original draft
preparation, Visualization
Jusheng Gao: Resources, Data curation
Huaihai Chen: Formal analysis, Data curation, Visualization
Zeyuan Zhang: Investigation, Resources
Jing Huang: Resources
Lefu Lv: Investigation
Jinfang Tan: Conceptualization, Supervision, Project administration
Xiaoqian Jiang: Methodology, Writing-Reviewing and Editing, Project administration,
Funding acquisition



**Competing interests:**
The authors declare that they have no known competing financial interests or personal
relationships that could have appeared to influence the work reported in this paper.

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



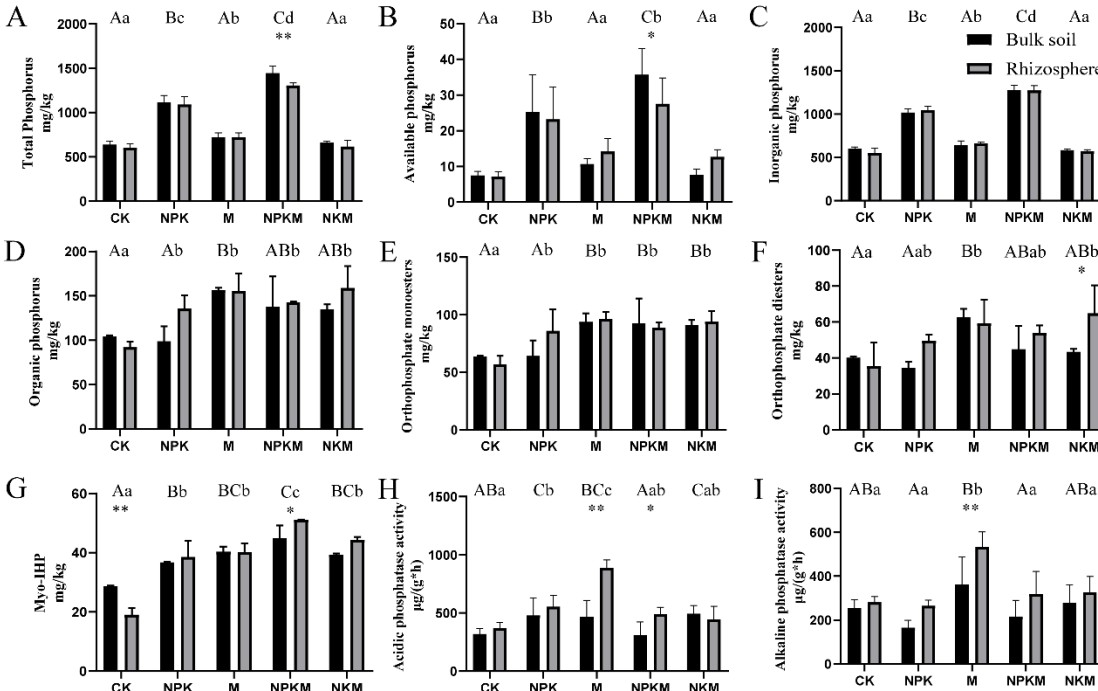

Fig. 1 Different phosphorus forms and phosphatase activities in five treatments (CK, NPK, M, NPKM, NKM) and

two sample types (rhizosphere and bulk soil), where A: Total phosphorus, B: Available phosphorus, C: Inorganic

phosphorus, D: Organic phosphorus, E: Orthophosphate monoesters, F: Orthophosphate diesters, G: Myo-IHP, H:

Acidic phosphatase activity, I: Alkaline phosphatase activity. Significant differences between treatments in bulk soil

are indicated by capital letters (p<0.05, n = 4). Significant differences between treatments in rhizosphere are indicated

by lowercase letters (p<0.05, n = 4). Significant differences between rhizosphere and bulk soil are indicated by

asterisks, where * p < 0.05, ** p < 0.01 (Duncan's test, n=4)



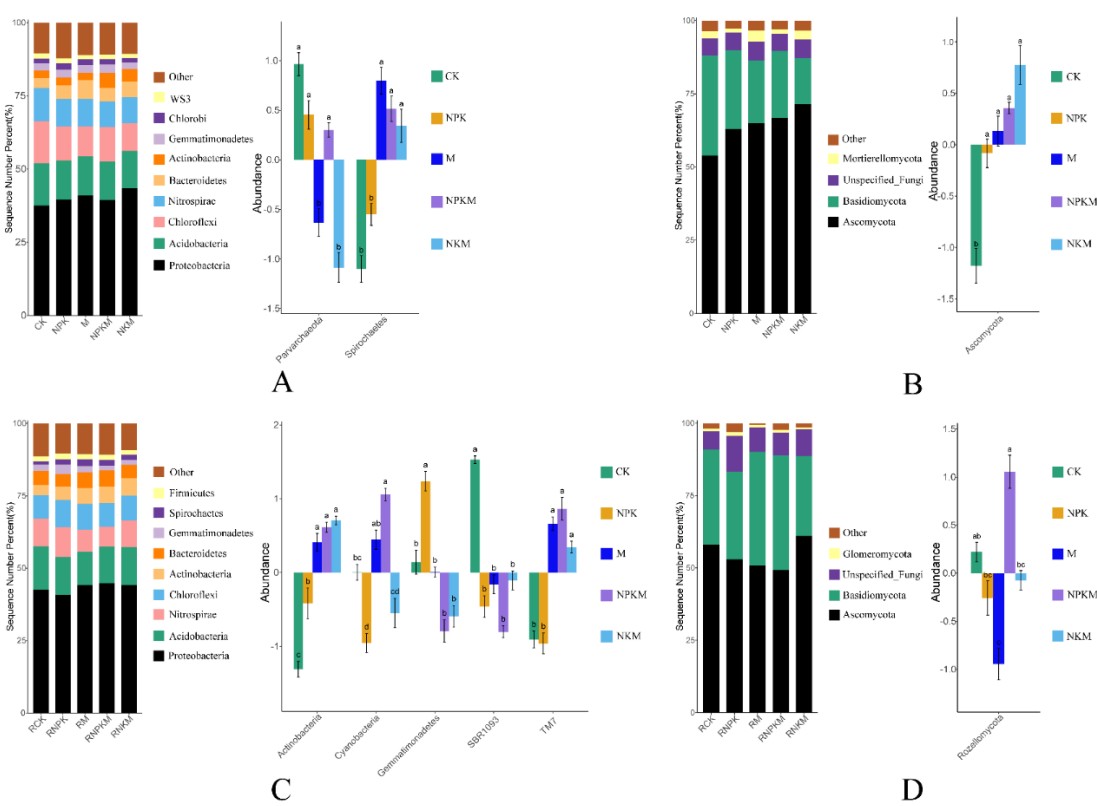

Fig. 2 The microbial relative abundance (left) and the features with significant differences (Anova + Duncan, p<0.05, n=6) between groups (right) at the phylum level in five treatments (CK, NPK, M, NPKM, NKM). Capital letters means different classification (A: bacteria in bulk soil, B: fungi in bulk soil, C: bacteria in rhizosphere soil, D: fungi in rhizosphere soil)



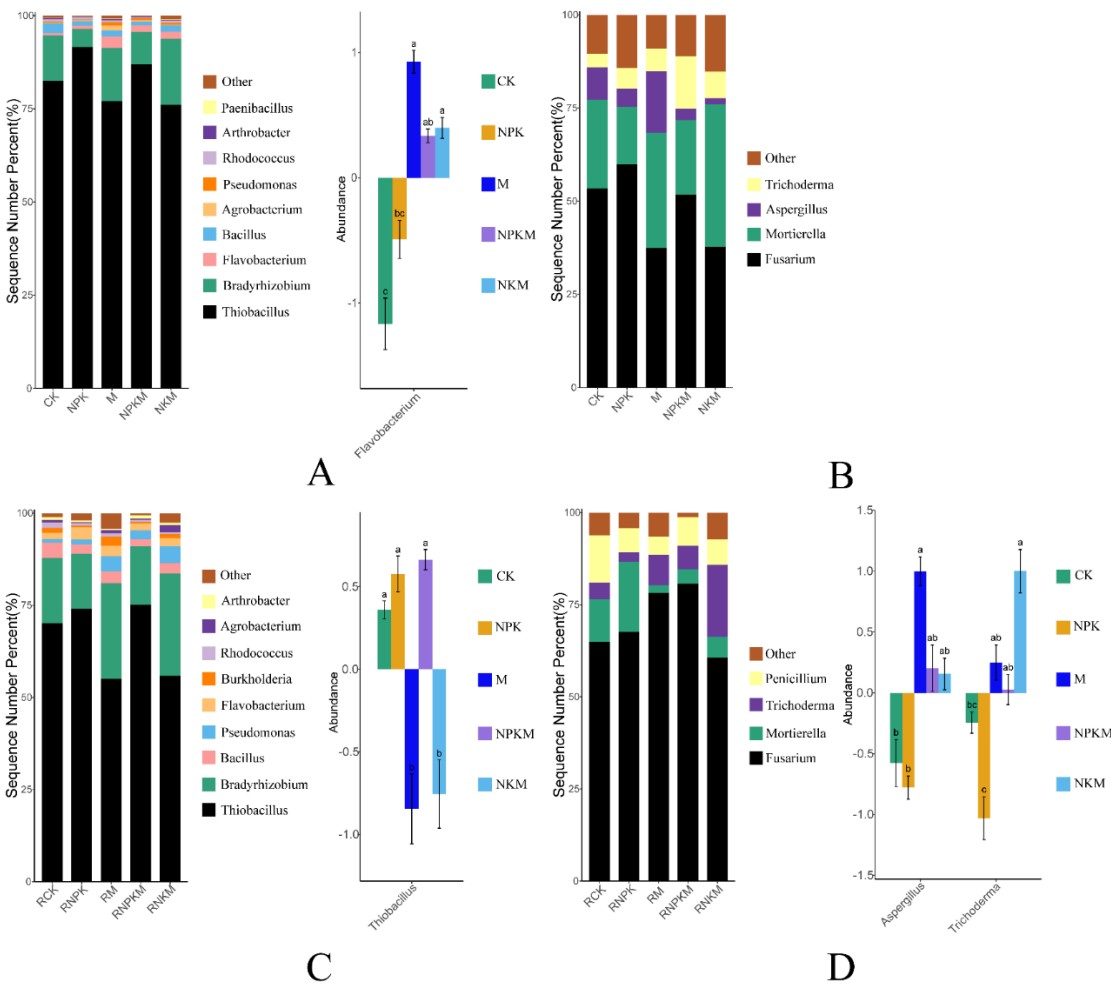

Fig. 3 The relative abundance of phosphorus-solubilizing microbe (left) and features with significant differences
(Anova + Duncan, p<0.05, n=6) between groups (right) at the genus level in five treatments (CK, NPK, M, NPKM,
NKM). Capital letters means different classification (A: bacteria in bulk soil, B: fungi in bulk soil, C: bacteria in
rhizosphere soil, D: fungi in rhizosphere soil)

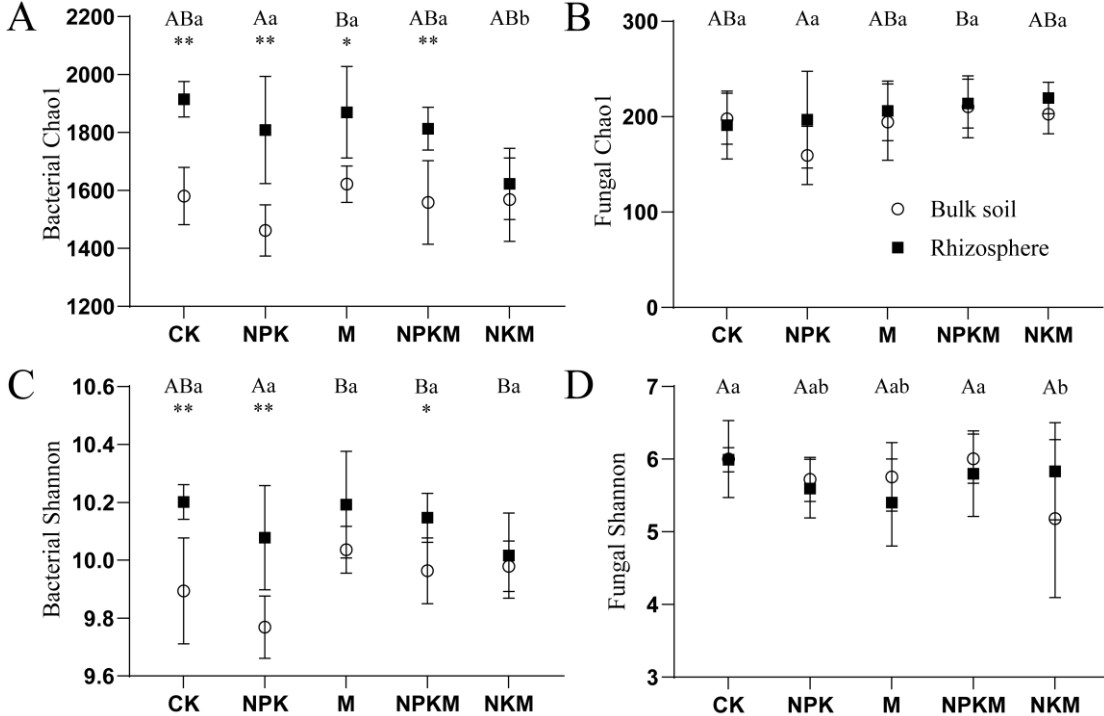


Fig. 4 Mean ± SE values for microbial α-diversity (A: Bacterial Chao1 index, B: Fungal Chao1 index, C: Bacterial Shannon index, D: Fungal Shannon index) in five treatments (CK, NPK, M, NPKM, NKM) and two sample types (rhizosphere and bulk soil). Significant differences between treatments in bulk soil are indicated by capital letters (p<0.05, n = 6). Significant differences between treatments in rhizosphere are indicated by lowercase letters (p<0.05, n = 6). Significant differences between rhizosphere and bulk soils are indicated by asterisks, where * p < 0.05, ** p < 0.01 (Duncan's test, n=6)




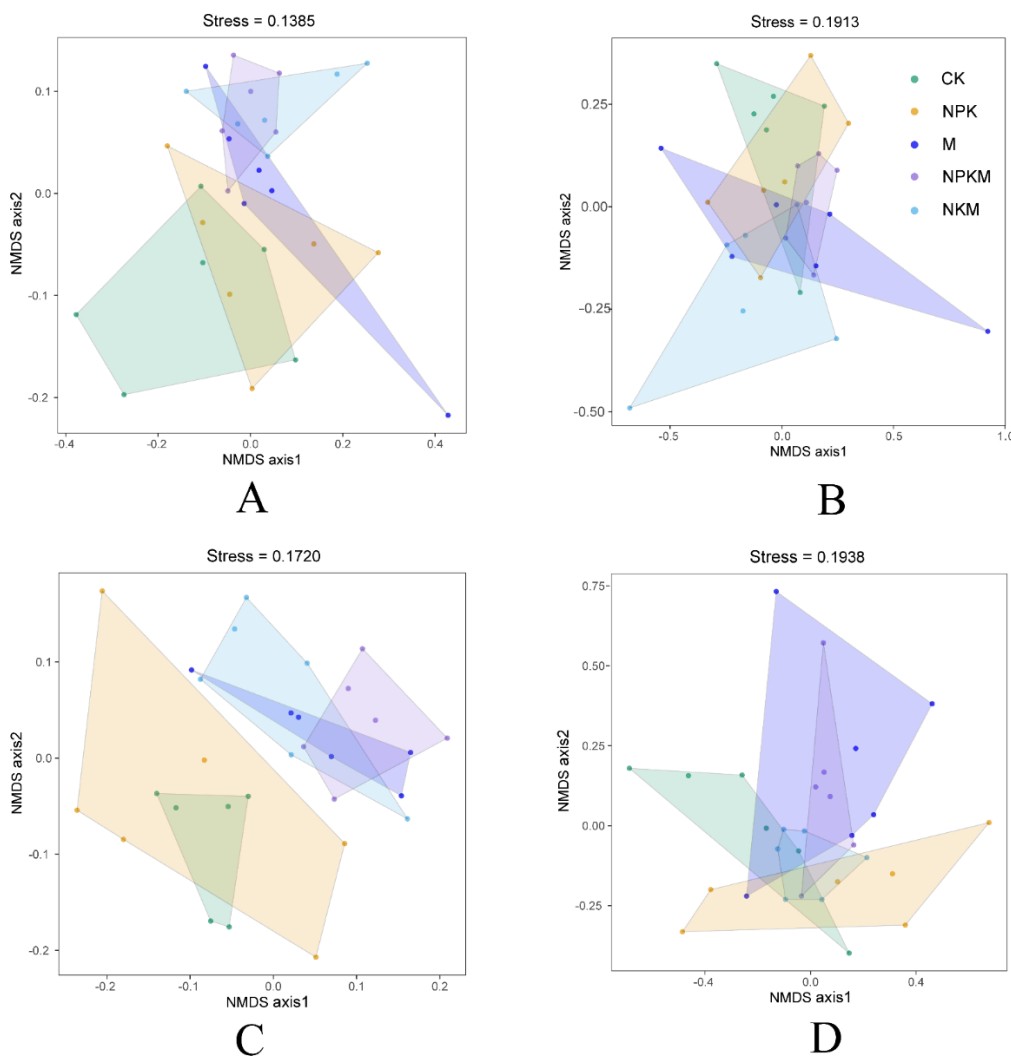

Fig. 5 Nonmetric multi-dimensional scaling (NMDS) ordination of the microbial community by comparing with Bray-Curtis distance similarities based on the abundance of OTUs. Capital letters means different classification (A: bacteria in bulk soil, B: fungi in bulk soil, C: bacteria in rhizosphere soil, D: fungi in rhizosphere soil)



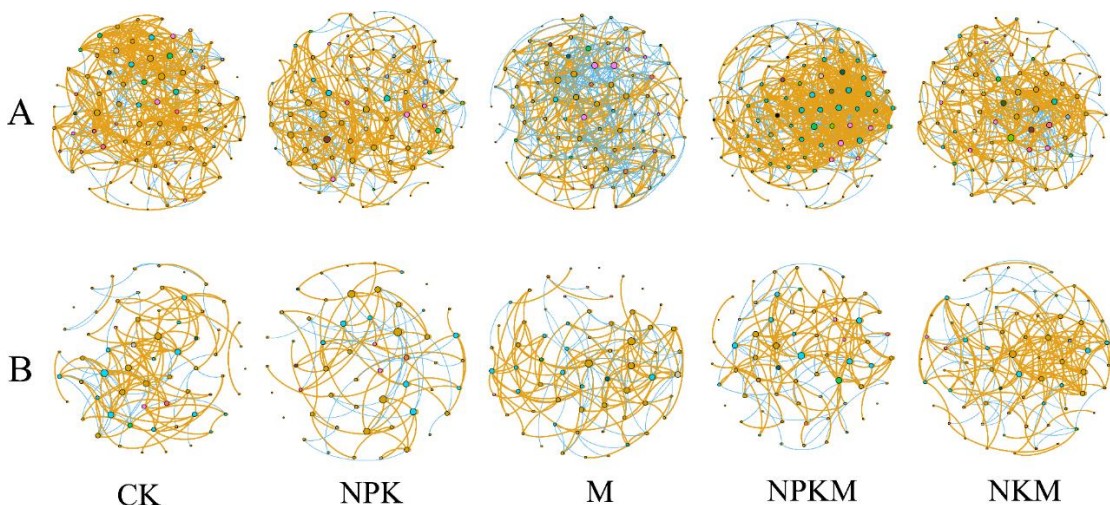

Fig. 6 Network of co-occurring bacterial (A) and fungal (B) OTUs across five fertilizer treatments. Only Spearman's correlation coefficient $r > 0.6$ or $r < -0.6$ significant at $P < 0.01$ is shown. The nodes are colored according to phylum. orange edges represent positive correlations and blue edges represent negative correlations. Node size presents the connecting numbers of each OUT.

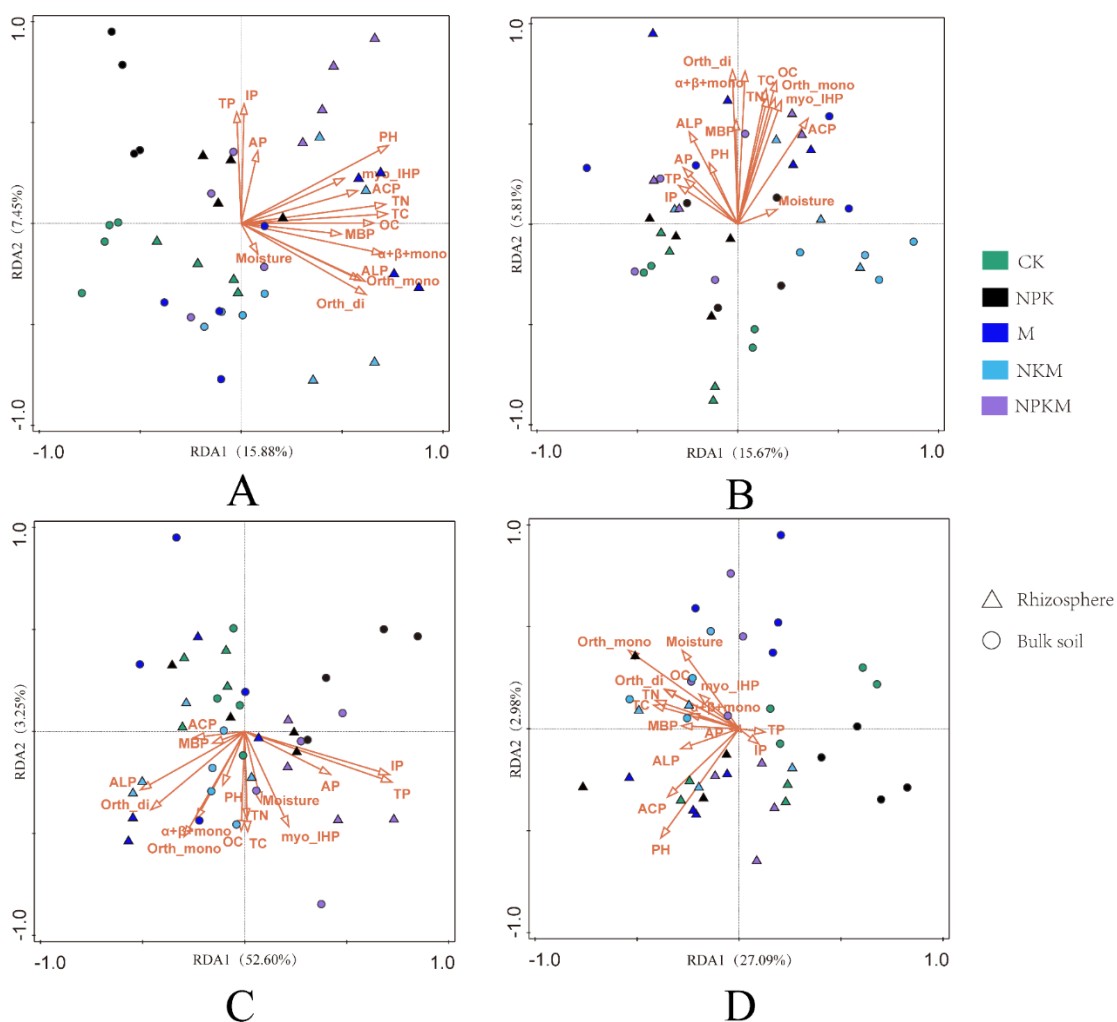

Fig. 7 Correlations between soil properties and the community structure of total bacteria (A), total fungi (B), phosphorus-solubilizing bacteria (C), and phosphorus-solubilizing fungi (D) as determined by redundancy analysis (RDA). MBP, microbial biomass phosphorus; TP, total phosphorus; IP, inorganic phosphorus; AP, available phosphorus; Orth-mono, orthophosphate monoester; Orth-di, orthophosphate diesters; Myo-IHP, myo-Inositol hexakisphosphate; α+β+mono, α- and β-glycerophosphates and mononucleotides; ACP, activity of acidic phosphatase; ALP, activity of alkaline phosphatase.



Table 1
The soil properties in five treatments (CK, NPK, M, NPKM, NKM) and two sample types (Bulk and Rhizosphere
soil).

| Soil properties | Sample type | CK | NPK | M | NPKM | NKM |
|---|---|---|---|---|---|---|
| Total C (g/kg) | Bulk soil | 19.08±0.26 a** | 23.38±0.56 b** | 30.30±0.23 d** | 34.08±0.22 e | 28.80±0.87 c |
| | Rhizosphere | 20.93±0.56 a | 26.38±1.59 b | 33.63±0.81 d | 34.48±0.15 d | 30.18±0.29 c |
| Organic C (g/kg) | Bulk soil | 15.13±0.30 a | 17.88±1.16 b | 23.43±1.42 c | 26.18±0.68 d | 22.35±0.37 c |
| | Rhizosphere | 16.38±0.66 a | 18.90±1.00 b | 25.33±1.69 d | 25.88±1.14 d | 22.23±0.52 c |
| Total N (g/kg) | Bulk soil | 2.25±0.06 a | 2.58±0.05 b** | 3.28±0.10 c* | 3.53±0.10 d | 3.00±0.12 e* |
| | Rhizosphere | 2.35±0.06 a | 2.93±0.15 b | 3.45±0.06 d | 3.58±0.05 d | 3.25±0.06 c |
| C/N | Bulk soil | 8.48±0.10 a* | 9.08±0.35 b | 9.26±0.29 b | 9.67±0.23 c | 9.60±0.08 c |
| | Rhizosphere | 8.90±0.06 a | 9.02±0.19 ab | 9.75±0.09 c | 9.64±0.11 c | 9.29±0.25 b |
| pH | Bulk soil | 5.84±0.08 ab | 5.85±0.02 ab | 5.89±0.17 ab | 5.89±0.01 b | 5.76±0.05 a |
| | Rhizosphere | 5.95±0.06 a | 6.13±0.02 b | 6.15±0.09 b | 6.19±0.15 b | 6.07±0.04 b |
| Gravimetric Moisture | Bulk soil | 0.41±0.05 a | 0.43±0.00 a | 0.45±0.03 ab | 0.49±0.03 b | 0.48±0.02 b |
| | Rhizosphere | 0.39±0.03 a | 0.41±0.03 ab | 0.43±0.02 ab | 0.44±0.03 b | 0.44±0.03 b |
| Nitrate-N (mg/kg) | Bulk soil | 0.60±0.05 a | 0.66±0.23 a** | 1.30±0.90 a | 0.99±0.26 a | 1.45±1.13 a |
| | Rhizosphere | 0.95±0.36 a | 0.80±0.34 b | 1.42±0.62 a | 1.26±0.61 a | 1.34±0.62 a |
| Ammonia-N (mg/kg) | Bulk soil | 12.15±2.92 a | 11.08±2.27 a | 10.40±2.32 a | 8.66±1.46 a | 11.82±3.24 a |
| | Rhizosphere | 10.07±2.59 a | 17.23±1.02 a | 11.79±1.60 a | 11.38±2.21 a | 11.66±3.90 a |
| Microbial biomass P (mg/kg) | Bulk soil | 3.70±3.49 a | 7.95±5.70 ab | 15.56±8.42 ab | 12.39±9.60 b | 10.45±3.83 ab |
| | Rhizosphere | 5.53±2.71 a | 12.59±8.06 ab | 17.90±4.27 b | 21.40±8.59 b | 17.77±11.14 b |

Values are means ± standard error.
Significant differences between treatments are indicated by lowercase letters ($p<0.05$, n = 4).
Significant differences between rhizosphere and bulk soil are indicated by asterisks, where * $p < 0.05$, ** $p < 0.01$
(Duncan's test, n=4)













Table 2
The phosphorus species in five treatments (CK, NPK, M, NPKM, NKM) and two sample types (Bulk and
Rhizosphere soil).

| P Form or Compound Class | Sample type | CK | NPK | M | NPKM | NKM |
|---|---|---|---|---|---|---|
| NaOH-EDTA extracted phosphorus mg/kg | Bulk soil | 253.86±34.05 a | 560.13±22.78 b | 361.80±2.00 a | 738.70±40.05 c | 304.31±1.66 a |
|  | Rhizosphere | 243.29±38.26 a | 546.67±101.12 b | 345.77±38.23 a | 685.59±36.28 c | 348.40±52.34 a |
| Orthophosphate mg/kg | Bulk soil | 146.84±32.01 a | 459.07±2.71 b | 202.19±5.91 a | 598.10±1.61 c | 167.24±2.98 a |
|  | Rhizosphere | 148.23±33.88 a | 409.37±88.42 b | 187.37±20.00 a | 540.29±40.60 c | 186.66±25.94 a |
| Pyrophosphate mg/kg | Bulk soil | 2.94±0.64 a | 2.30±3.26 a | 3.01±1.34 a | 3.00±4.24 a | 2.52±1.23 a |
|  | Rhizosphere | 2.73±1.42 a | 1.73±2.45 a | 2.74±1.02 a | 2.56±3.62 a | 2.89±1.71 a |
| Orthophosphate monoesters mg/kg | Bulk soil | 63.87±0.75 a | 64.31±13.36 a | 93.87±7.26 b | 92.73±21.40 b | 91.09±4.29 b |
|  | Rhizosphere | 56.90±7.41 a | 85.97±18.57 b | 96.28±6.33 b | 88.72±4.76 b | 93.99±9.14 b |
| Myo-IHP mg/kg | Bulk soil | 28.69±0.17 a** | 36.73±0.22 b | 40.40±1.68 bc | 44.86±4.35 c* | 39.29±0.48 bc |
|  | Rhizosphere | 19.03±2.31 a | 38.58±5.51 b | 40.21±2.98 b | 51.18±0.04 c | 44.43±0.95 b |
| Scyllo-IHP mg/kg | Bulk soil | 5.03±0.08 a | 6.90±3.29 a | 10.15±3.16 a | 8.98±4.25 a | 7.52±1.05 a |
|  | Rhizosphere | 4.45±1.02 a | 8.19±1.77 a | 9.37±1.00 a | 7.96±3.21 a | 8.12±2.79 a |
| Other monoesters mg/kg | Bulk soil | 30.16±0.49 ab | 20.69±9.86 a | 43.33±8.74 b | 38.89±12.79 ab | 44.29±2.76 b |
|  | Rhizosphere | 33.45±10.73 a | 39.20±11.29 a | 46.70±2.35 a | 29.57±1.59 a | 41.43±10.99 a |
| Orthophosphate diesters mg/kg | Bulk soil | 40.21±0.66 a | 34.44±3.45 a | 62.72±4.69 b | 44.87±12.81 ab | 43.46±1.59 ab* |
|  | Rhizosphere | 35.43±13.20 a | 49.61±3.42 ab | 59.38±12.93 b | 54.03±4.06 a | 64.86±15.55 b |
| DNA mg/kg | Bulk soil | 15.31±2.32 ab | 6.90±3.29 a* | 22.20±2.21 b | 11.97±8.49 ab | 13.38±0.24 ab |
|  | Rhizosphere | 12.34±6.90 a | 21.58±3.82 a | 18.15±8.52 a | 13.65±4.84 a | 20.06±9.32 a |
| α+β+mono mg/kg | Bulk soil | 24.90±1.67 a | 27.54±0.16 a | 40.52±6.90 b | 32.90±4.32 a | 30.08±1.83 a** |
|  | Rhizosphere | 23.10±6.30 a | 28.03±0.40 a | 41.22±4.40 b | 40.38±0.78 b | 44.80±6.23 b |

Myo-IHP: myo-Inositol hexakisphosphate; Scyllo-IHP: Scyllo-Inositol hexakisphosphate; α+β+mono, α- and β-
glycerophosphates and mononucleotides; Values are means ± standard error.
Significant differences between treatments are indicated by lowercase letters (p<0.05, n = 2). Significant differences
between rhizosphere and bulk soil are indicated by asterisks, where * p < 0.05, ** p < 0.01 (Duncan's test, n=2)
