# Peer review of "The role of long-term mineral and manure"

_EGUsphere, 2022_

## Referee Comment (RC1)

Comments

This study not only investigated P species but also studied the relevant enzymes and microbes responsible for P transformation. It provides valuable information for understanding P cycling based on long-term fertilization management with different treatments. Thus, I would like to recommend it for publication after resolving the following problems:

1. The hypothesis seemed to have no meaning. A different response is obvious, but what specific difference should be given? Line 83-85. Please revise accordingly.

2. More information about the experimental field and design should be given since it is a long-term experiment (i.e., 38 years). Also, the previous studies involved in this research area should be properly cited. Line 93-99.

3. So, the author only conducted sampling once? The sampling details should be given. Line 102.

4. How many hours were used for determining moisture? Line 109-110

5. The pretreatment method for OC determination should be given. Line 110.

6. The solid-liquid ratio of soil extracts should be presented in line 112.

7. The author should explain why they determine acid and alkaline phosphatase activity in this study. Also, what kind of phosphatase is produced by microbes? Line 124-126.

8. In section 3 Results, there are many citations of others' studies. I think

it is better to describe the result of this study, while the comparison or explanation of this study should be presented in the Discussion. Please check Section 3 thoroughly and make this section clear and concise.

9. "bacteria" in line 372 should be "bacterial".

10. Although the author proposed hypotheses in the introduction, they did not answer the hypothesis. In the Discussion, the author should mention whether they achieved the goal through this study.

11. Conclusion should be more concise.

---

## Author Comment (AC1)

**Comments to the Author:**

This study not only investigated P species but also studied the relevant enzymes and microbes responsible for P transformation. It provides valuable information for understanding P cycling based on long-term fertilization management with different treatments. Thus, I would like to recommend it for publication after resolving the following problems:

**Response:** *Thanks for your constructive suggestions on our paper. We have revised the paper according to your suggestions. The following is the answers and revisions we have made in response to the questions and suggestions on an item-by-item basis. A detailed explanation of the revision follows below.*

**Comment # 1:**

The hypothesis seemed to have no meaning. A different response is obvious, but what specific difference should be given? Line 83-85. Please revise accordingly.

**Response:** *Revised as suggested.*

*Line 85-89:We hypothesized that (1) long-term input of inorganic fertilizers accumulate more inorganic P but the manure application and rhizosphere may accelerate the accumulation of organic P and (2) the long-term manure fertilization and rhizosphere could accumulate more organic nutrients, thus driving the separation of bacterial communities compared to the mineral fertilizer application.*

**Comment # 2:**

More information about the experimental field and design should be given since it is a long-term experiment (i.e., 38 years). Also, the previous studies involved in this research area should be properly cited. Line 93-99.

**Response:** *As the reviewer suggested, more information about the experimental field and design has been given, and the previous studies involved in this research area have also been properly cited in Line 96-100.*

*Line 96-100:Rice (Oryza sativa) is the major crop in this region. The early rice was transplanted at the end of April and harvested in July, and the late rice was transplanted at the end of July and harvested in October. All straw (except the rice stubble) was removed from the fields after each seasonal rice harvest (Zhang et al., 2017; Yang et al., 2012).*

**Comment # 3:**

So, the author only conducted sampling once? The sampling details should be given. Line 102.

**Response:** *As the reviewer suggested, we have added more sampling details in Line 108-113.*

*Line 108-113: Bulk soil samples collection with five different fertilizer treatments were conducted before the harvest of late rice in October 2020 with field replications. In each field, three soil cores (0-20 cm topsoils) were collected and then pooled to form a composite sample. Besides, before the rhizosphere soil collection, the bulk soil was manually removed and approximately 1 mm of soil on the rice roots was collected as rhizosphere soil (Shao et al., 2021).*

**Comment # 4:**

How many hours were used for determining moisture? Line 109-110

**Response:** *The time was 16 h and we have added it in the Line 118-119.*

*Line 118-119: Soil moist content was measured by drying moist soil at 105 °C for 16 h until it became a constant mass.*

**Comment # 5:**

The pretreatment method for OC determination should be given. Line 110.

**Response:** *We have added this information in Line 119-122.*

*Line 119-122: Total carbon (TC), organic carbon (OC), and total nitrogen (TN) were determined by CHNS elemental analyzer (Vario EL Cube manufactured by Elementar, Germany) (Schumacher, 2002). The soil was pretreated by 1M HCl with soil-liquid ratio of 1:1 before OC determination.*

**Comment # 6:**

The solid-liquid ratio of soil extracts should be presented in line 112.

**Response:** *As suggested, we have added this information in Line 122-125.*

*Line 122-125: 1g soil was extracted with 5 mL KCl (2M) to determine for ammonia-N ($NH_4^+$) by indophenol blue colorimetric method (Dorich and Nelson, 1983), and for nitrate-N ($NO_3^-$) by dual-wavelength ultraviolet spectrophotometry (Norman et al., 1985).*

**Comment # 7:**

The author should explain why they determine acid and alkaline phosphatase activity in this study. Also, what kind of phosphatase is produced by microbes? Line 124-126.

**Response:** *Thanks for the useful suggestion. We have added this information in Line*

*134-140.*

*Line 134-140: Additionally, phosphatases could mediate soil P transformation and recycling. The alkaline phosphatase in soil is released by bacteria, whereas acid phosphatase can derive from plants, fungi and bacteria (Nannipieri et al., 2011; Acosta-Martínez and Ali Tabatabai, 2011). The activities of acid and alkaline phosphatase were indicators to reflect the microbial activity and P cycling ability in soil, and were assayed by the method described by Tabatabai and Bremner (1969) using p-nitrophenyl phosphate as substrate at 37 °C.*

**Comment # 8:**

In section 3 Results, there are many citations of others' studies. I think it is better to describe the result of this study, while the comparison or explanation of this study should be presented in the Discussion. Please check Section 3 thoroughly and make this section clear and concise.

**Response:** *Revised as suggested. The explanation in the Result section was presented in the Discussion section.*

**Comment # 9:**

"bacteria" in line 372 should be "bacterial".

**Response:** *Revised as suggested.*

**Comment # 10:**

Q: Although the author proposed hypotheses in the introduction, they did not answer the hypothesis. In the Discussion, the author should mention whether they achieved the goal through this study.

**Response:** *Thanks for your constructive suggestion. Now the hypotheses have been answered in the Discussion.*

*Line 325-327: The application of inorganic P fertilizer mainly increased the concentration of inorganic P but manure fertilization accelerated the accumulation of organic P in soil, which was consistent with our hypotheses (Fig. 1C and D).*
*Line 358-361: On the other side, long-term organic fertilization did not change the bacterial richness and evenness, and even promoted the separation of bacterial communities. This conclusion was also expected in our hypothesis.*

**Comment # 11:**

Conclusion should be more concise.

**Response:** *Revised as suggested.*

*Line 432-444: Long-term inorganic and organic fertilization managements brought different effects on P accumulation, microbial community, and PSB. Long-term mineral fertilization increased inorganic and available P concentrations, while manure fertilization increased soil organic P concentrations, microbial biomass P contents, and potential organic P mineralization. The turnover of P by bacteria seems strong under long-term organic fertilization and rhizosphere soil considering that more organic nutrient was provided for bacteria and the bacterial community diversity increased. Furthermore, inorganic P fertilization increased the abundance of Thiobacillus whereas organic fertilization increased the abundance of Flavobacterium, Aspergillus, and Trichoderma. The concentrations of TP and IP strongly influenced by inorganic P fertilization were key factors driving the diversity of soil PSB community. These findings provide useful insights into P accumulation, turnover, and soil P sustainable fertility under different fertilization strategies.*

---

## Author Comment (AC2)

**Comments to the Author:**

This study aims to examine 38-year fertilization experiments under 5 fertilizer treatments were conducted to determine their effects on P pool accumulation, soil microbial communities, and phosphate solubilizing microorganisms (PSM) in paddy soils. Authors claimed that different fertilizer management could affect P species, wherein inorganic fertilizer treatments increased inorganic and available P concentrations. However, organic fertilizer treatments increased organic P concentrations, microbial biomass P contents, and alkaline phosphatase activity. Additionally, this study demonstrated that the compositions of PSM also related to different fertilizer managements. Inorganic fertilization increased the abundance of Thiobacillus whereas organic fertilization raised the abundance of Flavobacterium, Aspergillus, and Trichoderma. This study provided sufficient and sophisticated data. However, some technical errors such as some unit formats should modify throughout this manuscript. This manuscript also needs to be edited for the sentence construction. Collectively, I recommend the publication of this study after a minor revision. Few specific comments were given below.

**Response:** *Thanks for your constructive suggestions on our paper. We have revised the paper according to your suggestions. A detailed explanation of the revision follows below.*

**Comment # 1:**

Abstract: The scientific significant should mentioned in the abstract rather than merely state the results. Abstract should emphasize the significance of the work and state the purpose, the main findings of this work.

**Response:** *Thanks for your suggestion. We have revised the Abstract accordingly.*

*Line 21-23:Understanding soil P transformation and turnover under various fertilization managements is important for evaluating sustainable P fertility and potential bioavailability in agriculture managements.*

*Line 40-42:These findings are beneficial for understanding the variation of inorganic and organic P pool, and microbial community especially for PSM under long-term inorganic and/or organic fertilization.*

**Comment # 2:**

Line 26-27: Please change mg/L to mg $L^{-1}$.

**Response:** *Revised as suggested.*

**Comment # 3:**

Materials and methods: Line 101: 0-20 cm?

**Response:** *Yes, the sampling depth is 0-20 cm. We have revised it in the manuscript.*

**Comment # 4:**

Line 103-104: Please provide detail methods for collecting the rhizosphere soil.

**Response:** *We have provided detail methods for collecting the rhizosphere soil in Line 111-113.*

*Line 111-113: Besides, before the rhizosphere soil collection, the bulk soil was manually removed, and approximately 1 mm of soil on the rice roots was collected as rhizosphere soil (Shao et al., 2021).*

**Comment # 5:**

Line 110-111: Please provide methods for soil organic carbon measurement.

**Response:** *Revised as suggested in Line 119-122.*

*Line 119-122: Total carbon (TC), organic carbon (OC), and total nitrogen (TN) were determined by CHNS elemental analyzer (Vario EL Cube manufactured by Elementar, Germany) (Schumacher, 2002). The soil was pretreated by 1M HCl with soil-liquid ratio of 1:1 before OC determination.*

**Comment # 6:**

Line 117-118: Please confirm the extraction method of acidic soil available P was referred to the Bray No.1.

**Response:** *The extraction method of available P in this study was referred to Olsen method, because according to the previous studies, the Olsen method could be useful in both acidic and alkaline soil, and was able to explain P status in acidic soil (Shao et al., 2017; Jordan-Meille et al., 2012).*

**Comment # 7:**

Results:

Line 209, 216, and 217: mg kg$^{-1}$.

**Response:** *Revised as suggested.*

**Comment # 8:**

Discussion:

I suggest authors could provide important diagrammatic sketch for this study.

**Response:** *Thanks for your suggestion. We have added a diagrammatic sketch as suggested.*

[Figure]

*Fig. 8 A diagrammatic sketch showing different responses of P accumulation, soil microbial communities and the PSM after long-term mineral or manure fertilization. ↑, increase; -, no effect; ↓, decrease.*

**Reference**

Jordan-Meille, L., Rubæk, G. H., Ehlert, P. A. I., Genot, V., Hofman, G., Goulding, K., Recknagel, J., Provolo, G., and Barraclough, P.: An overview of fertilizer-P recommendations in Europe: soil testing, calibration and fertilizer recommendations, Soil Use and Management, 28, 419-435, https://doi.org/10.1111/j.1475-2743.2012.00453.x, 2012.

Shao, X., Zhang, J., and Guo, L.: Assessing the Feasibility of Olsen P and P Sorption Parameters in Acidic Soils, Communications in Soil Science and Plant Analysis, 48, 2049-2060, https://doi.org/10.1080/00103624.2017.1406101, 2017.